# Differences in perceived popularity and social preference between bullying roles and class norms

Eva M. Romera[1]*, Ana Bravo[1], Rosario Ortega-Ruiz[1], René Veenstra[2]

**1** Psychology Department, Universidad de Córdoba, Córdoba, Spain, **2** Sociology Department, University of Groningen, Groningen, Netherlands

☯ These authors contributed equally to this work.
* eva.romera@uco.es

**Data Availability Statement:** The fully anonymized data are all contained within the Supporting Information file.

## Abstract

The aim of this study was to examine differences in perceived popularity and social preference of bullying roles and class norms. In total, 1,339 students (48% girls) participated: 674 primary school ($M$ = 10.41 years, $SD$ = 0.49) and 685 secondary school students ($M$ = 12.67 years, $SD$ = 0.80). Peer nominations and perceptions of class norms were collected. The results showed the highest perceived popularity among aggressors and defenders, except in anti-bullying primary school classes, where aggressors had low levels of popularity. In pro-bullying secondary school classes school, female victims had the lowest popularity levels. These findings suggest that class norms and personal variables as gender and school levels are important to understand bullying roles. Practical implications are discussed to guide teachers and practitioners according to the importance to adapt antibullying programs to the characteristics of the group in each school level and gender.

## Introduction

Bullying can be defined as a pattern of intentional aggressive behavior which is repeated over time and aimed at one or more victims by one or more aggressors, who assume a role of superiority over the former, in other words, an 'imbalance of power' [1]. This behavior involves the whole group and many students play a role in the bullying process [2,3], which may lead to the practice being perpetuated and even accepted [4]. Previous research has examined the relation between bullying roles and popularity and social preference [5,6]. But little is known about the extent to which the relation between bullying roles and popularity and social preference depends on whether a class has a pro- or anti-bullying norm [7,8]. The aim of our study is to examine differences in popularity and social preference assigned to each role—victim, aggressor, defender, or outsider—depending on type of norms accepted by the class.

### Sociometrical studies in bullying

According to the Goal-Framing Theory [9], people process information, define situations, and act according to factors that favor or hinder the fulfilment of their social objectives. Previous

**Funding:** This work was supported by the I+D+i, Ministerio de Industria, Economía y Competitividad Spain, Grant: project PSI2016-74871-R (www. mineco.gob.es) to EMR. The funder had no role in study design, data collection and analysis, decision to publish, or preparation of the manuscript.

**Competing interests:** The authors have declared that no competing interests exist.

research has highlighted that popularity and social preference are the two main social goals during childhood and adolescence [10]. Popularity resembles the level of visibility, prestige and power [5]. Social preference refers to the maintenance of close, friendly relationships with peers within the group [6]. It likely that bullies' are oriented to achieving domination, not to the question whether or not they are socially preferred by others [11]. However, the bully will not needlessly sacrifice social preference, which opens the door for norm influence on bullying.

It is well known that levels of perceived popularity and social preference in students involved in bullying depend on the role that they play in these violent acts. Aggressors have moderate to high levels of rejection by some peers, and medium to high levels of acceptance by others [6,12]. This duality may be due to the performance of a bi-strategic control strategy [13]. Research showed that aggressors seek to improve their popularity and maintain positive affective relationships with peers who are important to them, while intimidating and dominating others [7,14,15]. Victims have, on average, low perceived popularity and social preference levels. Defenders have high popularity and social preference levels [6]. Meanwhile, students who remain outsiders to bullying situations tend to have low popularity levels and low to medium acceptance levels [6,16]. So far, only a few studies have examined perceived popularity and social preference levels of bullying roles (referring to aggressors, victims, defenders and outsiders) comparatively.

## Pro- and anti-bullying class norms

Bullying can be seen as an expression of a particular relational group dynamic by all participants, not just by an aggressor and a particular victim. The norm of the group has been recognized as a main characteristic of the class network. These group norms are taken as a model for personal behavior, having an impact on students' attitudes and decisions [17]. Recent studies have stressed that group norms influence children's psychological, emotional, and moral attitudes to bullying [18] and what type of feedback peers give when bullying occurs [8].

Research into group norms has highlighted the different features that are naturally formed in a class. In our study, two types of norms will be identified: anti- and pro-bullying norms [19]. In classes with anti-bullying norms, students perceived negative consequences of bullying behaviors (mocking a classmate, taking part in the bullying, laughing with others), but positive consequences of antibullying behaviors (befriending with the victim, telling the teacher about the bullying). In pro-bullying groups, students perceived positive consequences of bullying behaviors [19,20]. Most research has analyzed what is the effect of high or low bullying acceptance norms, finding that in classes with high bullying acceptance (pro-bullying norms) both students with high rejection and low popularity levels and students with high popularity levels and popularity motivation develop more often aggressive behaviors [8]. In addition, aggressive behaviors are more accepted by peers, who perceive them as a way to promote one's social position (popularity and social preference) [21]. To date, there is little research which describes the extent to which defender behavior depends on anti- and pro-bullying class norms. A recent study highlighted, however, how difficult it is for students to defend others in pro-bullying classes, because in such classes it is difficult to show affective empathy, which is needed to defend victims [21].

## Bullying roles, perceived popularity and social preference between educational level and gender

There is much research that has noted differences in popularity and social preference and the prevalence of bullying roles between primary and secondary schools [6], and between boys

and girls [22]. With respect to perceived popularity and social preference levels, aggressors and defenders often present similar popularity levels during primary school, but aggressors appear to become more popular than defenders at the beginning of secondary school [23], due to the increase in the acceptance of and a predisposition toward aggressive behavior in early adolescence [24]. Victim's levels of popularity and social preference are lower in adolescence than in childhood [6]. With respect to gender, some researchers did not find developmental differences [3], and others noted only in girls an increase with age in the number of aggressors and only in boys a decrease with age in the number of defenders [6]. Further research into this topic is necessary.

Gender research indicates that aggressive behavior correlated most with high popularity levels for boys and low popularity levels for girls, and vice versa for defending behaviors [22]. However, little is known about these characteristics in relation to class norms. Thus, an additional focus of our research is to examine whether perceived popularity and social preference levels of bullying roles are different between primary and secondary school and between boys and girls. Gender differences are expected according to previous research that shows that females and males differ in their social preferences in many dimensions, including altruism [25] and honesty [26]. As well as, whether group norms are linked to these differences.

## The present study

It has been shown that bullying roles differ in perceived popularity and social preference. These differences in status can vary depending on class norms. The norms established in the group are essential for understanding how the social network is structured within the classes. It would be interesting to know how bullying classroom norms are related with bullying roles, popularity and social preference. Our study aims were: 1) to examine differences between bullying roles in perceived popularity and social preference, 2) to analyze if these differences were the same in groups with anti- or pro-bullying norms, and 3) analyze if the relation between perceived popularity, social preference, bullying roles and class norms differed between primary and secondary education and between boys and girls.

Taking these objectives into account, and based on the findings of previous studies, we put forward four hypotheses. First, we expected that aggressors and defenders would have high levels of popularity, defenders would have the highest levels of social preference and victims the lowest. Second, we expected that aggressors would have higher levels of social preference and popularity in pro-bullying classes than in anti-bullying classes, and that defenders would have lower levels of social preference and popularity in pro-bullying classes than in anti-bullying classes. Third, defenders and aggressors would have similar popularity levels in primary school, but aggressor would have higher popularity levels than defenders in secondary school. Fourth, we expected that girls would have higher popularity levels as defenders and lower levels as aggressors, whereas boys would have higher popularity levels as aggressors.

## Method

### Participants

In our study 1,339 students (48% girls) aged between nine and 15 years old ($M = 11.53$; $SD = 1.35$), from 53 classes in 14 schools in southern Spain (28% urban and 72% rural) participated. The 674 preadolescents were in the last two years of primary school and were aged between 9 and 13 years old ($M = 10.41$; $SD = 0.49$), belonging to 30 classes. In each class, 13 to 27 students took part ($M = 22.50$; $SD = 4.11$). The 685 adolescents were in the first two years of secondary school and were aged between 11 and 15 years old ($M = 12.67$; $SD = 0.80$), belonging to 23 classes. In each class, 17–35 students took part in the study ($M = 28.65$; $SD = 4.32$).

## Measures

**Social dimensions.** This measurement was obtained using four sociometric questions: 'Which peers are popular?' and 'Which are not popular?' for perceived popularity, and 'Who do you like?' and 'Who do you dislike?' for social preference. Students could nominate an unlimited number of classmates (both boys and girls) for each question. To do this, students had to provide a number of their classmates, as written on the blackboard.

**Bullying roles.** The different roles were assigned from the students' answers to three questions: *Who bully others*, *who are victims*, and *who defend victims*? Bullying definition and examples were previously given to students. Number of nominations received for each student were counted and data were standardized per class. Students were assigned to a role if they obtained a scored above the classroom average on the scale for that role ($z > 0$). No role was assigned in cases when two roles had higher than average scores, but their difference was below 0.10 (for more information see 6). Four roles were established: aggressor, defender, victim, and outsider.

**Class norms.** Participants completed Perceived Group Norms questionnaire (PGN, 18). This questionnaire presents five situations about how each participant perceives what the response of their class group would be. They were asked to imagine what their class group would do if a classmate behaved in the following ways: (1) befriending a victim of bullying; (2) laughing with others when someone is being bullied; (3) telling the teacher about a bullying incident which has occurred; (4) taking part in bullying; (5) making others laugh by continually mocking a classmate.

They were offered eight response options and were asked to choose only one per situation. The answers chosen were categorized into three groups: (a) the option *nothing much would happen* was categorized as 0 in all situations; (b) the options: *others would think they are a good person*, *others would show them their approval* and *others would feel admiration for them* were categorized in Situations 1 & 3 as anti-bullying responses and in Situations 2, 4 & 5 as pro-bullying options; (c) the options: *others would begin to avoid them*, *others would think they are stupid*, and *others would show them that they disagree with them* were categorized in Situations 1, 2 & 3 as pro-bullying responses and in Situations 2, 4 & 5 as anti-bullying options. Option 8— *Would something else happen*? *If so*, *what*?—was used to elicit free answers, which were later categorized using the same criteria. Pro-bullying responses were categorized as 1 and anti-bullying responses as -1. Next, the five scores were added together and then divided by the total number of situations, giving an average score for each subject, where negative values nearer -1 indicated perceived anti-bullying attitudes from and positive values indicated perceived pro-bullying attitudes. This questionnaire was translated into Spanish by the method of parallel back translation.

## Procedure

The study used a convenience sample based on accessibility. The school heads were informed of the research objectives and were asked to participate. Families were asked for their written and signed consent. Families of 14 students (1%) did not give their consent. It was stressed that the study was voluntary and anonymous: in order to guarantee anonymity, participants had to name their peers using a number on a list given by the teacher. Data was collected during school hours in their usual classes. Out of the total number of participants, 132 students (9%) were absent when the data was collected. Only classes where more than 80% of the students attended were selected. The study was conducted in accordance with the Declaration of Helsinki, and the protocol was approved by the Ethics Committee for Bioethics and Biosafety at the University of Cordoba.

## Data analysis

To calculate sociometric variables, nominations given by students for each social dimension and for each class were put into a *directed adjacency matrix*, where values of 0 and 1 represented the absence and presence of nominations between two actors. Next, matrices were fed into the UCINET 6.85 sociometric data analysis program [27,28] and Freeman's degree centrality for each dimension was obtained for each actor [29], using standardized scores ranging from 0–100. These indices were transferred to a matrix in SPSS v.24, where perceived popularity (level of popularity minus level of unpopularity) and social preference levels (level of acceptance minus level of rejection) were calculated for each participant. *Class norms* were calculated from the average value per class. The values for each class were fed into a second matrices in the SPSS v.24. The type of norm of each class was assigned following two procedures: a) a K-means classification cluster, and b) based on whether the class average was above (pro-bullying class) or below (anti-bullying class) of the average of all participating classes. Both procedures made the same allocation of classes to each type of group.

A MANOVA analysis was carried out to find out whether there were any significant differences in dimensions of perceived popularity and social preference depending on bullying roles. Games-Howell and Bonferroni post hoc tests were used according to the homogeneity of variance. Cohen's *d* statistic [30] was used to estimate effect sizes for the differences between groups. A second MANOVA analysis was carried out to discover the interaction between social status, bullying roles, and class norms. An ANOVA was performed to identify roles and groups between which these differences occurred. To analyze these differences, a new variable was established called *roles according to class norms* and was assigned eight values, one for each role in each type of classes.

MANOVA and ANOVA tests were replicated to study the differences between primary and secondary schools. To find out how gender interacted with the social status levels of each role, one MANOVA was carried out to compare roles and gender in the total sample and another to compare roles and gender according to class norms. Both analyses were carried out for primary and secondary schools. Levels of significance of $p < .05$ were accepted in all analyses.

## Results

### Descriptive results

In primary school, 47% (14 classes) of the classes had an anti-bullying norm and 53% (16 classes) a pro-bullying norm. In secondary school, 52% (12 classes) of classes had anti-bullying norm and 48% (11 classes) pro-bullying norm. A total of 1,318 (missing 1.6%) participants were assigned to a bullying role. In the anti-bullying classes, 110 (8%) participants were assigned as aggressors, 121 (9%) as victims, 207 (16%) as defenders, and 245 (19%) as outsiders. In the pro-bullying classes, 122 (9%) were aggressors, 112 (8%) victims, 181 (14%) defenders, and 220 (17%) outsiders.

Differences were only found in the prevalence of bullying roles according to gender ($\chi^2 =$ 69.44; $p < .001$). Boys were more often aggressors (74%) and girls' defenders (58%). There were no differences between educational level.

### Characteristics of popularity and social preference in bullying roles

Our findings on the relation between bullying roles and popularity and social preference showed significant differences (see Table 1). The Games-Howell post hoc tests showed differences in popularity levels between all bullying roles: victims ($M$ = -14.23; $SD$ = 24.60), outsiders ($M$ = -3.11; $SD$ = 16.82), aggressors ($M$ = 5.05; $SD$ = 22.82), and defenders ($M$ = 8.98;

**Table 1. Differences in bullying roles, popularity and social preference.**

| Popularity | | | | | | | | | |
|---|---|---|---|---|---|---|---|---|---|
| | Between Groups | | | MANOVA | | Two-by-two comparison (Games Howell) | | | Cohen's d |
| | N | M | SD | F(df) | p | | Mean Difference | p | |
| O | 464 | -3.11 | 16.82 | 71.7 (3) | < .001 | O-2 | -8.16 | < .001 | 0.43[a] |
| A | 232 | 5.05 | 22.82 | | | O-3 | 11.12 | < .001 | 0.56[b] |
| V | 233 | -14.23 | 24.60 | | | O-D | -12.08 | < .001 | 0.66[b] |
| D | 389 | 8.98 | 19.73 | | | A-V | 19.28 | < .001 | 0.81[c] |
| Total | 1318 | -0.07 | 21.91 | | | A-D | -3.93 | .131 | 0.19[a] |
| | | | | | | V-D | -23.21 | < .001 | 1.07[c] |

| Social Preference | | | | | | | | | |
|---|---|---|---|---|---|---|---|---|---|
| | Between Groups | | | MANOVA | | Two-by-two comparison (Games Howell) | | | Cohen's d |
| | N | M | SD | F(df) | p | | Mean Difference | p | |
| O | 464 | 41.56 | 20.69 | 44.2 (3) | < .001 | O-A | 7.94 | < .001 | 0.35[a] |
| A | 232 | 33.62 | 26.62 | | | O-V | 12.43 | < .001 | 0.58[b] |
| V | 233 | 29.13 | 23.38 | | | O-D | -6.46 | < .001 | 0.33[a] |
| D | 389 | 48.02 | 18.78 | | | A-V | 4.49 | .271 | 0.18[a] |
| Total | 1318 | 39.87 | 22.87 | | | A-D | -14.40 | < .001 | 0.65[b] |
| | | | | | | V-D | -18.89 | < .001 | 0.92[c] |

| Popularity: primary school | | | | | | | | | |
|---|---|---|---|---|---|---|---|---|---|
| | Between Groups | | | ANOVA | | Two-by-two comparison (Games Howell) | | | Cohen's d |
| | N | M | SD | F(df) | p | | Mean Difference | p | |
| O | 234 | -4.16 | 16.11 | 41.5 (3) | < .001 | O-A | -8.94 | .004 | 0.47[a] |
| A | 110 | 4.78 | 24.51 | | | O-V | 6.17 | .025 | 0.34[a] |
| V | 130 | -10.33 | 21.64 | | | O-D | -16.29 | < .001 | 0.95[c] |
| D | 184 | 12.13 | 18.38 | | | A-V | 15.10 | < .001 | 0.66[b] |
| Total | 658 | 0.53 | 21.20 | | | A-D | -7.35 | .036 | 0.35[a] |
| | | | | | | V-D | -22.46 | < .001 | 1.14[c] |

| Popularity: secondary school | | | | | | | | | |
|---|---|---|---|---|---|---|---|---|---|
| | Between Groups | | | ANOVA | | Two-by-two comparison (Games Howell) | | | Cohen's d |
| | N | M | SD | F(df) | p | | Mean Difference | p | |
| O | 230 | -2.04 | 17.49 | 37.8 (3) | < .001 | O-V | -7.33 | .007 | 0.39[a] |
| A | 122 | 5.29 | 21.40 | | | O-V | 17.12 | < .001 | 0.82[c] |
| V | 103 | -19.17 | 27.20 | | | O-D | -8.19 | < .001 | 0.43[a] |
| D | 205 | 6.15 | 20.50 | | | A-V | 24.45 | < .001 | 1.01[c] |
| Total | 660 | -0.82 | 22.59 | | | A-D | -0.86 | .985 | 0.04[a] |
| | | | | | | V-D | -25.31 | < .001 | 1.11[c] |

O. Outsider; A. Aggressor; V. Victim; D. Defender

[a] Low sample size effect (< .50);

[b] medium (.50–.80);

[c] high (> .80)

$SD = 19.73$), except between aggressors and defenders. Post hoc tests showed differences in social preference between all bullying roles: victims ($M = 29.13$; $SD = 23.38$), aggressors ($M = 33.62$; $SD = 26.62$), outsiders ($M = 41.56$; $SD = 20.69$), and defenders ($M = 48.02$; $SD = 18.78$), except between aggressors and victims.

MANOVA results in Table 1 showed significant differences only in popularity levels in the interaction between bullying roles and primary and secondary schools, $F(3, 1318) = 5.3$; $p < .001$. ANOVA separate analyses were performed for primary and secondary schools. The Games-Howell post hoc test showed significant differences in primary school between all roles, with the highest levels for defenders ($M = 12.13$; $SD = 18.38$), followed by aggressors ($M = 4.78$; $SD = 24.51$), outsiders ($M = -4.16$; $SD = 16.11$), and victims ($M = -10.33$; $SD = 21.64$). In secondary school, defenders ($M = 6.15$; $SD = 20.50$) and aggressors ($M = 5.29$; $SD = 21.40$) did no longer differ in popularity and scored higher than outsiders ($M = -2.04$; $SD = 17.49$) and victims ($M = -19.17$; $SD = 27.20$).

## Characteristics of the popularity and social preference of bullying roles according to class norms

MANOVA results also showed significant differences in the interaction between bullying roles and class norms for perceived popularity, $F(7, 1318) = 11.8$; $p < .001$, and social preference, $F(7, 1318) = 2.6$; $p = .047$. Table 2 shows our findings of the ANOVA for differences between roles according to class norms and social status. The Games-Howell post hoc test showed significant differences in popularity and social preference levels for each role. In pro-bullying classes, popularity levels were highest for both defenders ($M = 12.38$; $SD = 20.32$) and aggressors ($M = 7.38$; $SD = 23.22$), followed by outsiders ($M = -2.15$; $SD = 17.66$) and then victims ($M = -20.70$; $SD = 23.73$). In the social preference dimension, higher levels were found for defenders ($M = 49.89$; $SD = 18.89$), then outsiders ($M = 43.04$; $SD = 20.89$) and aggressors ($M = 36.63$; $SD = 26.30$), followed by victims ($M = 26.81$; $SD = 24.15$). In the anti-bullying classes, defenders ($M = 6.10$; $SD = 18.76$) and aggressors ($M = 2.46$; $SD = 22.32$) had higher levels of perceived popularity, followed by outsiders ($M = -4.00$; $SD = 15.99$) and victims ($M = -8.25$; $SD = 23.95$). In social preference, higher levels were found for defenders ($M = 46.89$; $SD = 18.62$) then outsiders ($M = 40.24$; $SD = 20.41$), followed by both victims ($M = 31.28$; $SD = 22.53$) and aggressors ($M = 30.27$; $SD = 26.70$).

MANOVA analysis in Table 2 showed significant differences only in popularity levels in the interaction between bullying roles and primary and secondary schools, $F(7, 1318) = 2.4$; $p = .017$. In primary school, in classes with pro-bullying norm, levels were higher for defenders ($M = 14.12$; $SD = 17.77$) and aggressors ($M = 8.52$; $SD = 22.33$) than outsiders ($M = -3.12$; $SD = 16.09$) and victims ($M = -16.01$; $SD = 20.59$). In classes with an anti-bullying norm, levels were higher only for defenders ($M = 10.06$; $SD = 18.87$) than victims ($M = -5.14$; $SD = 21.41$) and outsiders ($M = -5.15$; $16.13$). In secondary school, in classes with a pro-bullying norm, levels were higher for defenders ($M = 10.50$; $SD = 22.72$) and aggressors ($M = 6.41$; $SD = 24.08$) than victims ($M = -26.51$; $SD = 26.18$). In classes with an anti-bullying norm, levels were higher for aggressors ($M = 3.97$; $SD = 17.86$) and defenders ($M = 3.05$; $SD = 18.18$) than victims ($M = -12.23$; $SD = 26.53$).

## Characteristics of popularity and social preference according to gender

No differences were found for the sample as a whole in our findings of MANOVA test between bullying roles and gender, nor in MANOVA performed for the interaction between roles according to class norms and gender. In primary school, no differences were found in either of these two interactions. In secondary school, however, significant differences were found in MANOVA results of the interaction between roles according to class norms and gender, in particular for the dimension of perceived popularity, $F(7, 656) = 4.6$; $p < .001$.

To find out which roles according to class norms showed differences in popularity levels in boys and girls, Table 3 shows the results of two ANOVA tests. The Games-Howell post hoc

**Table 2. Differences in bullying roles and popularity and social preference according to class norms.**

### Popularity

| | Between Groups | | | MANOVA | | Two-by-two comparison (Games Howell) | | | |
|---|---|---|---|---|---|---|---|---|---|
| | N | M | SD | F(df) | p | | Mean Difference | p | Cohen's d |
| PO | 220 | -2.15 | 17.66 | 36.8 (7) | < .001 | PO-PA | -9.53 | .003 | 0.48[b] |
| AO | 245 | -4.00 | 15.99 | | | PO-PV | 18.54 | < .001 | 0.93[c] |
| PA | 122 | 7.38 | 23.22 | | | PO-PD | -14.53 | < .001 | 0.77[b] |
| AA | 110 | 2.46 | 22.32 | | | AO-AD | -10.09 | < .001 | 0.58[b] |
| PV | 112 | -20.70 | 23.73 | | | PA-PV | 28.08 | < .001 | 1.20[c] |
| AV | 121 | -8.25 | 23.95 | | | AA-AV | 10.71 | .012 | 0.46[a] |
| PD | 181 | 12.38 | 20.32 | | | PV-AV | -12.45 | .002 | 0.52[b] |
| PD | 207 | 6.10 | 18.76 | | | PV-PD | -33.08 | < .001 | 1.53[c] |
| Total | 1318 | -0.07 | 21.91 | | | AV-AD | -14.35 | < .001 | 0.69[b] |
| | | | | | | PD-AD | 6.28 | .037 | 0.32[a] |

### Social Preference

| | Between Groups | | | MANOVA | | Two-by-two comparison (Games Howell) | | | |
|---|---|---|---|---|---|---|---|---|---|
| | N | M | SD | F(df) | p | | Mean Difference | p | Cohen's d |
| PO | 220 | 43.04 | 20.89 | 20.7 (7) | < .001 | PO-PV | 16.22 | < .001 | 0.74[c] |
| AO | 245 | 40.24 | 20.41 | | | PO-PD | -6.85 | .014 | 0.34[a] |
| PA | 122 | 36.63 | 26.30 | | | AO-AA | 9.97 | .014 | 0.44[b] |
| AA | 110 | 30.27 | 26.70 | | | AO-AV | 8.97 | .007 | 0.43[a] |
| PV | 112 | 26.81 | 24.15 | | | AO-AD | -6.17 | .019 | 0.34[a] |
| AV | 121 | 31.28 | 22.53 | | | PA-PD | -13.26 | < .001 | 0.60[b] |
| PD | 181 | 49.89 | 18.89 | | | AA-AD | -16.14 | < .001 | 0.77[c] |
| PD | 207 | 46.89 | 18.62 | | | PV-PD | -23.08 | < .001 | 1.10[c] |
| Total | 1318 | 39.87 | 22.86 | | | AV-AD | -15.14 | < .001 | 0.78[b] |

### Popularity: primary school

| | Between Groups | | | ANOVA | | Two-by-two comparison (Games Howell) | | | Cohen's d |
|---|---|---|---|---|---|---|---|---|---|
| | N | M | SD | F(df) | p | | Mean Difference | p | |
| PO | 114 | -3.12 | 16.09 | 20.6 (7) | < .001 | | | | |
| AO | 120 | -5.15 | 16.13 | | | PO-PA | -11.64 | .017 | 0.64[b] |
| PA | 56 | 8.52 | 22.33 | | | PO-PV | 12.89 | .001 | 0.90[c] |
| AA | 54 | 0.90 | 26.23 | | | PO-PD | -17.24 | < .001 | 1.03[c] |
| PV | 62 | -16.01 | 20.59 | | | AO-AD | -15.20 | < .001 | 0.88[c] |
| AV | 68 | -5.14 | 21.41 | | | PA-PV | 24.53 | < .001 | 1.15[c] |
| PD | 94 | 14.12 | 17.77 | | | PV-PD | -30.12 | < .001 | 1.60[c] |
| PD | 90 | 10.06 | 18.87 | | | AV-AD | -15.20 | < .001 | 0.76[b] |
| Total | 658 | 0.67 | 21.20 | | | | | | |

### Popularity: secondary school

| | Between Groups | | | ANOVA | | Two-by-two comparison (Games Howell) | | | Cohen's d |
|---|---|---|---|---|---|---|---|---|---|
| | N | M | SD | F(df) | p | | Mean Difference | p | |
| PO | 106 | -1.12 | 18.24 | 19.5 (7) | < .001 | | | | |
| AO | 125 | -2.89 | 15.84 | | | PO-PV | 25.40 | < .001 | 1.21[c] |
| PA | 66 | 6.41 | 24.08 | | | PO-PD | -11.62 | .005 | 0.57[b] |
| AA | 56 | 3.97 | 17.86 | | | PA-PV | 32.92 | < .001 | 1.33[c] |
| PV | 50 | -26.51 | 26.18 | | | AA-AV | 16.21 | .008 | 0.73[b] |
| AV | 53 | -12.23 | 26.53 | | | PV-PD | -37.01 | < .001 | 1.55[c] |
| PD | 87 | 10.50 | 22.72 | | | AV-AD | -15.28 | .007 | 0.73[b] |
| AD | 117 | 3.05 | 18.18 | | | | | | |
| Total | 660 | -0.82 | 22.59 | | | | | | |

PO. Outsider (pro); AO. Outsider (anti); PA. Aggressor (pro); AA. Aggressor (anti); PV. Victim (pro); AV. Victim (anti); PD. Defender (pro); AD. Defender (anti)

[a] Effect of sample size low (< .50);

[b] medium (.50–.80);

[c] high (> .80)

**Table 3. Differences in bullying roles and perceived popularity to class norms and gender.**

| | | | | | | | | | |
|---|---|---|---|---|---|---|---|---|---|
| | **Popularity: Boys** | | | | | | | | |
| | **Between Groups** | | | **ANOVA** | | **Two-by-two comparison (Games Howell)** | | | Cohen's d |
| | N | M | SD | F(df) | p | | Mean Difference | p | |
| PO | 49 | -4.69 | 16.73 | 12.2 (7) | < .001 | | | | |
| AO | 59 | -3.22 | 15.87 | | | PO-PA | -13.20 | .044 | 0.66[a] |
| PA | 46 | 8.51 | 23.28 | | | AO-AV | 18.50 | .012 | 0.96[b] |
| AA | 43 | 6.07 | 17.78 | | | PA-PV | 28.25 | < .001 | 1.20[b] |
| PV | 30 | -19.74 | 24.56 | | | AA-AV | 27.80 | .001 | 1.33[b] |
| AV | 31 | -21.73 | 25.06 | | | PV-PD | -26.57 | < .001 | 1.24[b] |
| PD | 46 | 6.82 | 19.55 | | | VD-AD | -23.31 | .002 | 1.08[b] |
| PD | 41 | 1.59 | 19.29 | | | | | | |
| Total | 345 | -1.90 | 22.10 | | | | | | |
| | **Popularity: Girls** | | | | | | | | |
| | **Between Groups** | | | **ANOVA** | | **Two-by-two comparison (Games Howell)** | | | Cohen's d |
| | N | M | SD | F(df) | p | | Mean Difference | p | |
| PO | 57 | 1.95 | 20.81 | 12.2 (7) | < .001 | | | | |
| AO | 64 | -2.41 | 16.14 | | | | | | |
| PA | 19 | 1.84 | 26.45 | | | PO-PV | 38.62 | < .001 | 1.76[b] |
| AA | 13 | -2.97 | 16.99 | | | PA-PV | 38.50 | .001 | 1.51[b] |
| PV | 20 | -36.67 | 25.82 | | | PV-PD | -50.28 | < .001 | 2.03[b] |
| AV | 22 | 1.14 | 22.92 | | | PV-AV | -37.81 | < .001 | 1.59[b] |
| PD | 40 | 13.61 | 24.89 | | | | | | |
| PD | 76 | 3.84 | 17.62 | | | | | | |
| Total | 311 | .26 | 23.01 | | | | | | |

PO. Outsider (pro); AO. Outsider (anti); PA. Aggressor (pro); AA. Aggressor (anti); PV. Victim (pro); AV. Victim (anti); PD. Defender (pro); AD. Defender (anti).

[a] medium (.50–.80);

[b] high (> .80).

test showed significant differences in both genders. Boys in classes with a pro-bullying group had by far the lowest level of popularity when they were victims ($M$ = -19.74; $SD$ = 24.56), whereas aggressors ($M$ = 8.51; $SD$ = 23.28) scored higher than outsiders ($M$ = -3.22; $SD$ = 15.87). In classes with an anti-bullying norm, the only significant difference was between victims ($M$ = -21.73; $SD$ = 25.06) on the one hand and defenders ($M$ = 1.59; $SD$ = 19.29) and aggressors ($M$ = 6.07; $SD$ = 17.78) on the other hand. In girls, the only differences were found in pro-bullying classes where victims ($M$ = -36.67; $SD$ = 25.82) showed levels lower than all the other roles.

## Discussion

The aim of this study was to examine the sociometric characteristics of different bullying roles, taking into account the values of perceived popularity and social preference in relation to the type of class norms existing in primary and secondary school classes.

### Perceived popularity and social preference and bullying roles

Victims obtained the lowest levels in perceived popularity. This finding highlights the existence of an imbalance of power, where weaker students occupy the lowest ranking position in terms of popularity, and thus have a low social and group resource access [14]. In line with the

first hypothesis of this research, aggressors and defenders showed similar perceived popularity levels, indicating that both types of behavior are related to power, a high social position and better resource access within the group [14,24].

We expected to find that perceived popularity levels of aggressors were higher than defenders in secondary schools and that these levels were similar in primary schools. Nevertheless, we found that defenders and aggressors popularity levels was similarly high in secondary and different in primary schools, where defenders showed higher levels than aggressors. This finding is consistent with the developmental taxonomy model of antisocial behavior [31], by which aggressive and antisocial behavior is valued more positively at the beginning of secondary school [14,24]. In addition, these results highlight that, although adolescents value defending behavior in terms of prestige and social power, aggressive behaviors are equally popular at these ages. This similarity was not due to an increase in aggressors' popularity levels [14,24], but a fall in popularity levels of defenders. This could be because defending victims means facing up peers (aggressors and reinforcers) who may exhibit behavior accepted by the group during adolescence [32].

Regarding social preference, and in line with our first hypothesis defenders showed the highest levels, whereas victims and aggressors showed the lowest levels. In the case of aggressors, research has revealed high levels of both rejection and acceptance linked to that role [12]. However, in our study we observed a greater tendency to an attitude of rejection toward aggressors. It would be of interest in further research to examine who accepts and who rejects an aggressor, in order to broaden our knowledge about aggressors' social preference levels.

## Perceived popularity and social preference, bullying roles and class norms

With regard to perceived popularity, the type of class norm is related to victims' popularity levels, being lower in the pro-bullying group compared to the anti-bullying group. This result highlights the situation of loss of prestige to which victims are subjected when their own classmates take bullying and intimidation for granted as a normative group dynamic. Victims themselves occupy a disadvantaged social position, and the chances of breaking that dynamic or of being helped are therefore small [21]. Also, it was observed differences between pro and anti-bullying classes in the role of defenders, with higher values in the first ones. These differences between pro-bullying and anti-bullying classes can be due to hierarchical structures in popularity (asymmetries in students' popularity) [33]. In these types of pro-bullying groups, greater efforts are required on the part of educators to tackle bullying [34].

As for social preference, the situation of social disadvantage of victims depending of type of classes was less negative. Victims obtained similar levels than aggressors in both type of classes. Although previous studies reveal a relation between the degree of acceptance of aggressive behavior and pro-bullying norms [8,35], our results highlight that there are no differences between social preference and class norms.

Following the social-misfit model [36], those who deviate from the group norm tend to be rejected, while students support and even imitate the socially accepted group behavior. Thus, one might expect negative levels of social preference for the defender in the pro-bullying groups, precisely because they go against the class norms. However, in line with previous research focused on prosocial behaviors [35], we found that defenders, no matter the social norm, obtained the highest levels of social preference. This would be in the opposite direction with respect to our second hypothesis, and suggests that despite the type of social norm in a group, the moral values held by the individual nevertheless play an important role [18]. Our second hypothesis in relation with aggressors was also rejected, because aggressors did not stand out above the other roles for their high popularity and acceptance in the pro-bullying

groups. Further research is needed to enable us to understand why defending behavior is accepted, regardless of class norms. For this, the key could be the study of characteristics of a personal nature related to prosocial, empathic, assertive, and leadership behavior.

We are therefore faced with a type of anti-bullying group in which ethical considerations and a clear idea about what is right and wrong play an important role because the class norms dictate it, as opposed to the type of pro-bullying group in which the norms of the group enable behavior which is immoral to be seen as normal.

The differences between primary and secondary schools according to class norms were only relevant for the popularity dimension. Differences between two type of norms was only found in primary classes. Thus, aggressors in anti-bullying groups obtained low popularity levels, with no differences with victims, which may indicate a rejection of dominating behavior in these classes at these ages. This finding highlights the importance of knowing what norms are in use in the group from an early age, because these class norms influence on their personal evaluation about whether aggressive behaviors should be socially valued or not. The lack of differences between class norms in secondary schools, may be because during adolescence obtaining and maintaining high levels of popularity is more important than any other type of social goals [5,37], and aggression is a useful strategy to achieve it [14,24]. In addition, this result can help to explain the reduced effectiveness of anti-bullying programs in secondary schools [38,39]. This finding concerning aggressors is partially in line with our third hypothesis, because although they enjoyed greater popularity than the victim within their classes, these levels were not higher than those of defenders in pro-bullying classes.

Differences in status, bullying roles, and class norms by gender were only found in terms of popularity among secondary school students. Our findings did not support our hypothesis about that female defenders obtained higher popularity levels than female aggressors. This result brings up new questions about how aggressive girls are perceived nowadays within their classes, being able to expect a change in attitude (increased acceptance) toward aggressive behaviors among girls during adolescence. In addition, class norms played a key role in popularity levels of girls. In anti-bullying classes, there were no differences in popularity levels of girls between bullying roles. However, in pro-bullying classes, victims obtained significantly lower levels than other roles, including female victims in anti-bullying classes, who presented positive levels of popularity. This result is in line with previous research that emphasizes the close relation between isolation and victimization in girls [40], a situation which gets worse in classes where bullying is normative among peers. We therefore need to focus our attention on female victims in pro-bullying classes, who are vulnerable, and on consequences for their well-being and social adjustment.

For boys, no difference was observed between popularity and bullying roles according to class norms, with the levels of aggressors and defenders being higher than those of victims in both groups. This similarity between groups may be influenced by the natural characteristics of their biological development, where aggressive behavior tends to be seen as more socially acceptable [41]. However, the similarity between popularity levels of aggressors and defenders could highlight that defensive behaviors are also associated with high levels of popularity among boys. Further studies are required to delve deeper into differences between boys and girls in the network structure existing in the classes.

## Strengths, limitations and future lines of research

This study has its limitations. Both bullying roles and class norms cannot be static realities but should be approached from a dynamic and longitudinal perspective. This study should be

replicated with longitudinal data which would enable us to take into account changes, in line with previous longitudinal studies [10].

Despite these limitations, our study combines relational and contextual variables to characterize bullying roles and has set new research objectives. Future research should look into characteristics and composition of the micro-groups in the classes and examine perceived popularity and social preference at the dyadic level by answering research questions which will reveal which roles are nominated by whom in terms of social status.

## Conclusions

This study has shown that victims are rejected by classmates and can see that those who hurt and bully them enjoy the recognition of their peers. Thus, bullying prevention and intervention programs should be focused on encouraging the inclusion in the group of the more vulnerable and less popular students. Our findings reinforce the need to develop intervention programs different for primary and secondary education, especially with respect to change in popularity levels of aggressors and defenders, which was similar in secondary schools. As well as, class norms have more relevant role during primary school, being interesting to include the social context in prevention programs. Thus, primary students of anti-bullying classes have a worse perception of bullying behaviors than in pro-bullying classes. Lastly, prevention and intervention programs must be different in their gender focus issues, especially during secondary schools.

## Supporting information

**S1 Dataset.**
(SAV)

## Author Contributions

**Conceptualization:** Eva M. Romera, Rosario Ortega-Ruiz.

**Data curation:** Eva M. Romera.

**Formal analysis:** Eva M. Romera, Ana Bravo.

**Funding acquisition:** Eva M. Romera.

**Investigation:** Eva M. Romera, Ana Bravo, Rosario Ortega-Ruiz, René Veenstra.

**Methodology:** Eva M. Romera, Ana Bravo, Rosario Ortega-Ruiz, René Veenstra.

**Project administration:** Eva M. Romera.

**Resources:** Eva M. Romera, Rosario Ortega-Ruiz.

**Software:** Eva M. Romera.

**Supervision:** Eva M. Romera, Rosario Ortega-Ruiz, René Veenstra.

**Validation:** Eva M. Romera, Rosario Ortega-Ruiz.

**Visualization:** Eva M. Romera, Rosario Ortega-Ruiz.

**Writing – original draft:** Eva M. Romera, Ana Bravo, Rosario Ortega-Ruiz.

**Writing – review & editing:** Eva M. Romera, Ana Bravo, Rosario Ortega-Ruiz, René Veenstra.

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
