## [Decision Letter · Decision Letter 0]

10 Aug 2019

PONE-D-19-16704

Differences in perceived popularity and social preference between bullying roles and class norms: A study in primary and secondary education

PLOS ONE

Dear Dr Romera,

Thank you for submitting your manuscript to PLOS ONE. After careful consideration, we feel that it has merit but does not fully meet PLOS ONE’s publication criteria as it currently stands. Therefore, we invite you to submit a revised version of the manuscript that addresses the points raised during the review process.

Please find the reviewers' comments, as well as mine, below.

We would appreciate receiving your revised manuscript by Sep 24 2019 11:59PM. To enhance the reproducibility of your results, we recommend that if applicable you deposit your laboratory protocols in protocols.io, where a protocol can be assigned its own identifier (DOI) such that it can be cited independently in the future. For instructions see: http://journals.plos.org/plosone/s/submission-guidelines#loc-laboratory-protocols

We look forward to receiving your revised manuscript.

Kind regards,

Valerio Capraro

Academic Editor

PLOS ONE

Journal Requirements:

Additional Editor Comments (if provided):

I have now collected three reviews from three experts in the field. The reviewers are positive but suggest several revisions before publication. I have read the paper and I agree with their point of view. Therefore, I would like to invite you to revise your work, following their comments. Additionally, I would like to add one more comment. Reading the manuscript, I could not find the measure of social preference. What was this measure? Moreover, previous research shows that females and males differ in their social preferences in many dimensions, including altruism (https://link.springer.com/article/10.1007/s10683-011-9283-7, https://psycnet.apa.org/record/2016-09706-001, https://www.sciencedirect.com/science/article/pii/S0165176518301952), cooperation (https://europepmc.org/articles/pmc5909828), and honesty (http://journal.sjdm.org/18/18619a/jdm18619a.pdf, https://psycnet.apa.org/record/2018-66786-001). Since your work also focus on gender differences, I was wondering whether the fact that social preferences depend on the gender might affect your results in same ways.

Looking forward for the revision.

Reviewers' comments:

Reviewer's Responses to Questions

**Comments to the Author**

1. Is the manuscript technically sound, and do the data support the conclusions?

Reviewer #1: Yes

Reviewer #2: Yes

Reviewer #3: Yes

2. Has the statistical analysis been performed appropriately and rigorously? 

Reviewer #1: Yes

Reviewer #2: Yes

Reviewer #3: Yes

3. Have the authors made all data underlying the findings in their manuscript fully available?

Reviewer #1: Yes

Reviewer #2: No

Reviewer #3: Yes

4. Is the manuscript presented in an intelligible fashion and written in standard English?

Reviewer #1: Yes

Reviewer #2: Yes

Reviewer #3: Yes

5. Review Comments to the Author

Reviewer #1: In my view, the paper is original and contributes to the previous research with a significant novelty. The overall design of study is adequate. The manuscript is well-written, well-organized and easy to follow.

The research questions are clearly defined. The title and the abstract are clear and informative. The keywords are properly chosen.

The introduction summarizes the fundamentals, gives a critical evaluation of the previous research on the topic, and objectives. The variables being investigated are clearly identified and presented.

Participants studied are adequately described. Adequate methods to answer the research questions are used. The description of the methods is adequate.

The results answer the research questions and are well presented. The interpretation of the results and the discussion of the results are clear and derived from data.

The references are up to date and relevant.

My only suggestion for improving the paper: practical implications of the results should be discussed.

Reviewer #2: The presented work is a valuable and interesting study, correctly performed that will surely contribute to update the state of arts in this field. Nevertheless, some suggestions to improve the manuscript are worth to be commented (in chronological order):

Abstract

The Abstract would benefit from an inclusion of the rationale or practical implications of the results (see more comments below about this issue). By now, in the Abstract there is a long extension of specific results that do not offer a holistic or clear picture.

Introduction

Although the topic is really interesting, it is also true that a great amount of studies has already been performed in the last years around this topic. Therefore, a clear rationale for the need of this specific study is advisable. What is the original contribution of this study? This rationale may be focused on the preventive work or practical implications that could be extracted from the obtained results. This rationale may be added at the end of the Introduction section or in The present study section.

More information about the relation (similarities and differences) between perceived popularity and social preference would be appreciated. Both terms are defined in the text, but in an isolated way. By including their relation, the rationale of the study will surely improve. Why are these two concepts important and not for example, social impact, which is another sociometric index? Or is popularity the same concept than social impact? In this last case, the authors may justify why they adopt the term “popularity” instead of “social impact”.

In my personal opinion, the text would benefit from an extended explanation of anti- and pro-bullying norms, in a more specific way. By now, only the consequences of these type of norms into popularity and social preference are stated.

The lack of a clear rationale of the study is also evident in the paragraph of the objectives and hypothesis. All of them are correctly written, but not a specific connection between them can be found, that is, a common thread is needed. For example, why the sociodemographic variables (age and gender) are important in this context? To customize interventions or prevention programs?

Although the second objective of the study refers to all the possible differences between bullying roles in perceived popularity and social preference between groups with anti- or pro-bullying norms, the corresponding hypothesis only applies to aggressors.

Method

It would be advisable to include more information about the Social Dimension measure: gender-specific nomination or both genders nomination? How the nominations were processed? Were they standardized to allow comparison between classes? Or at least it would be great to indicate that more information about this issue will be found later in the Procedure section.

In the Bullying role measure, three questions are asked for three roles: aggressor, victim, and defender. But from which question do the “outsider role” emerges? In fact, it is not the absence of the other roles that defines the “outsider role”, but an explicit behavior of “doing nothing,” staying outside the bullying situations. This needs further explanation.

Another related thing would be asking about the rest of the existent bullying roles: reinforcer of the bully and assistant of the bully. Are they collapsed in the aggressor role or are not simply taken into account? The authors coherently include Salmivalli studies about bullying roles in the work. However, this author proposes six bullying roles. An explanation of why only four of them are chosen for this study is advisable.

Results

The tables would have to be in APA format that means deleting the shading parts.

Discussion

It would be very interesting and necessary to remember all the hypotheses as well as to include the explicit support or lack of support to these hypotheses.

What is the meaning of “hierarchical structures in popularity” in relation to the obtained result? I think the reader who does not know the concept (as well as me) would benefit from a brief explanation.

I do not exactly agree with the first limitation as it is written now: peer nominations are not designed to inform about the direct relation between different bullying roles or about how many, how often, or what type of bullying the roles have carried out. It would be more adequate to only state the convenience of comparing self-nominations with peer-nominations in future studies.

The practical implications of the findings for professionals (related to the rationale of the study quoted before) are not clearly developed in a specific way. This would be something very interesting to expand in the final section.

References

Some references present the initial letters in the title in capital letters while others not.

Reviewer #3: This work focuses on the problem of bullying from a very interesting perspective, with the aim of analyzing the degree of popularity and social preference in each of the different roles involved in the process of bullying between peers. In addition, the authors examine whether these differences in status depend on the social norms present in class, studying the influence of the existence of beliefs more related to aggressive behavior, or against it.

In general, from my point of view, the work meets the publication criteria for this journal. The manuscript is, in general terms, well articulated and written, the analyzes are pertinent and respond to the objectives of the study, and the findings represents an advance regarding the existing knowledge on the subject.

There are a number of minor issues that would be appropriate, however, for the authors to try to clarify. The sample of participating adolescents comes from 14 schools, of which 28% are of urban origin and 72% are rural. Could the authors comment if this fact poses a problem regarding the representativeness of the sample participating in the study? Have they considered carrying out some kind of analysis by type of educational center? Regarding the data collection instruments, could the authors specify the versions used and validated for the type of sample and language used? Finally, I believe that the Discussion section would benefit if it is articulated around the starting hypothesis.

6. PLOS authors have the option to publish the peer review history of their article (what does this mean?). If published, this will include your full peer review and any attached files.

Reviewer #1: Yes: David Álvarez-García

Reviewer #2: No

Reviewer #3: Yes: Estefanía Estévez

---

## [Author Response · Author response to Decision Letter 0]

20 Sep 2019

Dear editor, 

Attached please find the revised version of manuscript with a new title “Differences in perceived popularity and social preference between bullying roles and class norms.” On behalf of my co-authors, I would like to thank you and the three reviewers for the feedback on our manuscript. You asked that we address the concerns raised by yourself and the reviews. We made every effort to address each point raised in the revised manuscript and in the present letter.

Best wishes,

Dr. Eva Romera

Editor Comments 

 All relevant data are within the paper and its Supporting Information file.

 We have adapted the manuscript to the PLOS ONE style templates.

I could not find the measure of social preference. What was this measure? Moreover, previous research shows that females and males differ in their social preferences in many dimensions, including altruism (https://link.springer.com/article/10.1007/s10683-011-9283-7, https://psycnet.apa.org/record/2016-09706-001, https://www.sciencedirect.com/science/article/pii/S0165176518301952), cooperation (https://europepmc.org/articles/pmc5909828), and honesty (http://journal.sjdm.org/18/18619a/jdm18619a.pdf, https://psycnet.apa.org/record/2018-66786-001). 

Social preference was measured with these questions ‘Who do you like?’ and ‘Who do you dislike?’, according to previous sociometric studies.

Since your work also focus on gender differences, I was wondering whether the fact that social preferences depend on the gender might affect your results in same ways. 

We appreciate the suggestions about gender research according to different social preference measures. 

We have incorporated this in our manuscript to justify the need to examine gender differences.

Reviewer 1 

My only suggestion for improving the paper: practical implications of the results should be discussed. 

As suggested we have incorporated practical implications in the abstract and discussion section. 

Reviewer 2 

The presented work is a valuable and interesting study, correctly performed that will surely contribute to update the state of arts in this field. Nevertheless, some suggestions to improve the manuscript are worth to be commented (in chronological order) 

We thank your review and we value your comments to improve our manuscript.

Abstract 

The Abstract would benefit from an inclusion of the rationale or practical implications of the results (see more comments below about this issue). By now, in the Abstract there is a long extension of specific results that do not offer a holistic or clear picture. 

We have incorporated information about practical implications. We have summarized the main results of the study in two main ideas.

Introduction 

Although the topic is really interesting, it is also true that a great amount of studies has already been performed in the last years around this topic. Therefore, a clear rationale for the need of this specific study is advisable. What is the original contribution of this study? This rationale may be focused on the preventive work or practical implications that could be extracted from the obtained results. This rationale may be added at the end of the Introduction section or in The present study section. 

We have highlighted the need for this study: “It has been shown that bullying roles differ in perceived popularity and social preference. These differences in status can vary depending on class norms. The norms established in the group are essential for understanding how the social network is structured within the classes. It would be interesting to know how bullying classroom norms are related with bullying roles, popularity and social preference. Our study aims were: 1) to examine differences between bullying roles in perceived popularity and social preference, 2) to analyze if these differences were the same in groups with anti- or pro-bullying norms, and 3) analyze if the relation between perceived popularity, social preference, bullying roles and class norms differed between primary and secondary education and between boys and girls.” 

More information about the relation (similarities and differences) between perceived popularity and social preference would be appreciated. Both terms are defined in the text, but in an isolated way. By including their relation, the rationale of the study will surely improve. Why are these two concepts important and not for example, social impact, which is another sociometric index? Or is popularity the same concept than social impact? In this last case, the authors may justify why they adopt the term “popularity” instead of “social impact”. 

Previous research has highlighted that popularity and social preference are the two main social goals during childhood and adolescence (10). Popularity resembles the level of visibility, prestige and power (5). Social preference refers to the maintenance of close, friendly relationships with peers within the group (6). It likely that bullies’ are oriented to achieving domination, not to the question whether or not they are (11). However, the bully will not needlessly sacrifice social preference, which opens the door for norm influence on bullying.”

In my personal opinion, the text would benefit from an extended explanation of anti- and pro-bullying norms, in a more specific way. By now, only the consequences of these type of norms into popularity and social preference are stated. 

We have introduced the next sentences to explain norms: “In our study, two types of norms will be identified: anti- and pro-bullying norms (18). In classes with anti-bullying norms, students perceived negative consequences of bullying behaviors (mocking a classmate, taking part in the bullying, laughing with others), but positive consequences of antibullying behaviors (befriending with the victim, telling the teacher about the bullying). In pro-bullying groups, students perceived positive consequences of bullying behaviors (18,19).”

The lack of a clear rationale of the study is also evident in the paragraph of the objectives and hypothesis. All of them are correctly written, but not a specific connection between them can be found, that is, a common thread is needed. For example, why the sociodemographic variables (age and gender) are important in this context? To customize interventions or prevention programs? 

Gender and school level are relevant variables in the study of bullying, social status and class norms. It has been shown that there are important differences between boys and girls and older and younger students. Accordingly, we have included these sociodemographic variables. The observed differences in our study could guide the design of bullying prevention programs. It has been shown that the social relationships can vary depending on gender and school level. We have introduced a sentence at the end of the aims to highlight the relevance of study of these variables.

Although the second objective of the study refers to all the possible differences between bullying roles in perceived popularity and social preference between groups with anti- or pro-bullying norms, the corresponding hypothesis only applies to aggressors. 

We have introduced now a hypothesis with defenders.

Method 

It would be advisable to include more information about the Social Dimension measure: gender-specific nomination or both genders nomination? How the nominations were processed? Were they standardized to allow comparison between classes? Or at least it would be great to indicate that more information about this issue will be found later in the Procedure section. 

We have introduced more information about social dimension in Instrument section to clarify your questions. The rest of information is included in the Procedure section.

In the Bullying role measure, three questions are asked for three roles: aggressor, victim, and defender. But from which question do the “outsider role” emerges? In fact, it is not the absence of the other roles that defines the “outsider role”, but an explicit behavior of “doing nothing,” staying outside the bullying situations. This needs further explanation. Another related thing would be asking about the rest of the existent bullying roles: reinforcer of the bully and assistant of the bully. Are they collapsed in the aggressor role or are not simply taken into account? The authors coherently include Salmivalli studies about bullying roles in the work. However, this author proposes six bullying roles. An explanation of why only four of them are chosen for this study is advisable. Despite that Salmivalli’ studies had described six bullying roles, recent research has assigned bullies, assistants, and reinforcers to one combined bully/follower role. For that reason, this research has decided to ask students only about the three main bullying roles (aggressor, defender, and victim). This decision has been made according to the criteria used in recent studies (Pouwels et al., 2016, 2017). Also, the definition of the role of outsider is taken from these sociometric studies. We are not measuring if they do something or not. We only know if they are involved (as victim, defender or aggressor) or not (outsider).

Results 

The tables would have to be in APA format that means deleting the shading parts. 

We have adapted tables to APA style.

Discussion 

It would be very interesting and necessary to remember all the hypotheses as well as to include the explicit support or lack of support to these hypotheses. 

Thanks for your comment. We have mentioned the hypothesis in the Discussion section.

What is the meaning of “hierarchical structures in popularity” in relation to the obtained result? I think the reader who does not know the concept (as well as me) would benefit from a brief explanation. 

This expression refers to asymmetries in students’ popularity.

I do not exactly agree with the first limitation as it is written now: peer nominations are not designed to inform about the direct relation between different bullying roles or about how many, how often, or what type of bullying the roles have carried out. It would be more adequate to only state the convenience of comparing self-nominations with peer-nominations in future studies. 

Thank you very much for your suggestion. We have removed this limitation.

The practical implications of the findings for professionals (related to the rationale of the study quoted before) are not clearly developed in a specific way. This would be something very interesting to expand in the final section. We have introduced new ideas about practical implications according to the results.

References 

Some references present the initial letters in the title in capital letters while others not. 

We have corrected all the references.

Reviewer 3 

In general, from my point of view, the work meets the publication criteria for this journal. The manuscript is, in general terms, well articulated and written, the analyzes are pertinent and respond to the objectives of the study, and the findings represents an advance regarding the existing knowledge on the subject. 

Thank you very much for your review.

There are a number of minor issues that would be appropriate, however, for the authors to try to clarify. The sample of participating adolescents comes from 14 schools, of which 28% are of urban origin and 72% are rural. Could the authors comment if this fact poses a problem regarding the representativeness of the sample participating in the study? Have they considered carrying out some kind of analysis by type of educational center? The distribution of the schools in this study was done according to the characteristics of our province, with more rural than urban schools. Examining differences between rural and urban schools would be a topic for further research.

Regarding the data collection instruments, could the authors specify the versions used and validated for the type of sample and language used? 

Social dimension and bullying roles were sociometric questions. All of them were presented in Spanish. 

Class norm instrument was translated by the method of parallel back translation from English to Spanish. The values of the scale were calculated from the sum of the children’s answers in each bullying situation. This information has been included into the Instruments section.

Finally, I believe that the Discussion section would benefit if it is articulated around the starting hypothesis. 

We have restructured the Discussion section, following your suggestion.

---

## [Editor Report · Decision Letter 1]

24 Sep 2019

Differences in perceived popularity and social preference between bullying roles and class norms

PONE-D-19-16704R1

Dear Dr. Romera,

We are pleased to inform you that your manuscript has been judged scientifically suitable for publication and will be formally accepted for publication once it complies with all outstanding technical requirements.

With kind regards,

Valerio Capraro

Academic Editor

PLOS ONE
---

## [Editor Report · Acceptance letter]

3 Oct 2019

PONE-D-19-16704R1 

Differences in perceived popularity and social preference between bullying roles and class norms 

Dear Dr. Romera:

I am pleased to inform you that your manuscript has been deemed suitable for publication in PLOS ONE. Congratulations! Your manuscript is now with our production department. 

With kind regards,

on behalf of

Dr. Valerio Capraro 

Academic Editor

PLOS ONE